# Improving Mitochondrial Function in Skeletal Muscle Contributes to the Amelioration of Insulin Resistance by Nicotinamide Riboside

**DOI:** 10.3390/ijms241210015

**Published:** 2023-06-12

**Authors:** Qiuyan Li, Xuye Jiang, Yujia Zhou, Yingying Gu, Yijie Ding, Jing Luo, Nengzhi Pang, Yan Sun, Lei Pei, Jie Pan, Mengqi Gao, Sixi Ma, Ying Xiao, De Hu, Feilong Wu, Lili Yang

**Affiliations:** 1Guangdong Provincial Key Laboratory of Food, Nutrition and Health, Department of Nutrition, School of Public Health, Sun Yat-sen University, Guangzhou 510080, China; liqy99@mail2.sysu.edu.cn (Q.L.);; 2Foundation Center for Basic Metabolic Research, Section of Metabolic Genetics, Faculty of Health and Medical Sciences, University of Copenhagen, 1172 Copenhagen, Denmark

**Keywords:** insulin resistance, NAD, nicotinamide riboside (NR), AMPK activation, mitochondrial dysfunction, oxidative stress

## Abstract

High-fat diet (HFD)-induced insulin resistance (IR) in skeletal muscle is often accompanied by mitochondrial dysfunction and oxidative stress. Boosting nicotinamide adenine dinucleotide (NAD) using nicotinamide riboside (NR) can effectively decrease oxidative stress and increase mitochondrial function. However, whether NR can ameliorate IR in skeletal muscle is still inconclusive. We fed male C57BL/6J mice with an HFD (60% fat) ± 400 mg/kg·bw NR for 24 weeks. C2C12 myotube cells were treated with 0.25 mM palmitic acid (PA) ± 0.5 mM NR for 24 h. Indicators for IR and mitochondrial dysfunction were analyzed. NR treatment alleviated IR in HFD-fed mice with regard to improved glucose tolerance and a remarkable decrease in the levels of fasting blood glucose, fasting insulin and HOMA-IR index. NR-treated HFD-fed mice also showed improved metabolic status regarding a significant reduction in body weight and lipid contents in serum and the liver. NR activated AMPK in the skeletal muscle of HFD-fed mice and PA-treated C2C12 myotube cells and upregulated the expression of mitochondria-related transcriptional factors and coactivators, thereby improving mitochondrial function and alleviating oxidative stress. Upon inhibiting AMPK using Compound C, NR lost its ability in enhancing mitochondrial function and protection against IR induced by PA. In summary, improving mitochondrial function through the activation of AMPK pathway in skeletal muscle may play an important role in the amelioration of IR using NR.

## 1. Introduction

The prevalence of Type 2 diabetes mellitus (T2DM) has been increasing worldwide over the past few decades. It was estimated that approximately 537 million adults (aged 20–79) suffered from diabetes in 2021 [1]. On account of sedentary lifestyles and excessive calorie intake, more individuals have become obese. Obesity is a major contributor to insulin resistance (IR), which is one of the key pathophysiological processes involved in T2DM [2]. Skeletal muscle, liver and white adipose tissue are the key insulin-responding tissues. Among them, skeletal muscle accounts for 60–70% of the insulin-stimulating glucose uptake in the whole body [3]. Thus, IR in skeletal muscle takes a major responsibility for hyperglycemia [4]. Mitochondria are the center of the cellular energy metabolism that contributes to the regulation of oxidative stress, cellular redox balance and cellular signaling transduction. The abnormal function of mitochondria is associated with many diseases. Structural damage of mitochondria and a reduced number of mitochondria were universally found in skeletal muscle cells of IR patients [5,6,7]. Previous studies found that nutrient (especially lipid) overfeeding led to mitochondrial oxidative stress, which contributed to the development of IR. Increased mitochondrial reactive oxygen species (ROS) production and a decreased ratio of glutathione/glutathione disulfide (GSH/GSSG) were found in the skeletal muscle of both obese mice and individuals [8]. Reduction in mitochondrial ROS production was able to prevent IR in skeletal muscle [8,9,10].

Adenosine monophosphate-activated protein kinase (AMPK) is a crucial mediator regulating mitochondrial function and energy metabolism. Catabolic events mediated by AMPK include enhancing glucose uptake through phosphorylating TBC1 domain family member 1/4 (TBC1D1/4) and triggering glucose transporter type 4 (GLUT4) membrane translocation [11,12,13]. Previous studies showed that activation of AMPK could activate IRS1/AKT signaling pathway and increase skeletal muscle insulin sensitivity [14]. Meanwhile, AMPK can maintain mitochondrial function as it can directly or indirectly activate peroxisome proliferator-activated receptor gamma coactivator 1-alpha (PGC-1α) to increase mitochondrial biogenesis [15,16]. AMPK is also involved in the turnover of mitochondria via mitophagy, which helps to scavenge ROS produced by the damaged mitochondria [17].

Nicotinamide adenine dinucleotide (NAD) is an essential coenzyme of mitochondrial oxidative phosphorylation and energy metabolism [18]. Studies have shown that the levels of NAD in multiple tissues of the body drop significantly in individuals with obesity, T2DM and non-alcoholic fatty liver disease (NAFLD) [19]. Increasing NAD levels via NAD precursors can increase skeletal muscle insulin sensitivity in overweight or obese prediabetic women [20] and improve insulin sensitivity in diabetic mice [21]. Nicotinamide riboside (NR) is a key NAD precursor that can effectively increase NAD biosynthesis in the body [22]. Unlike nicotinic acid (NA) and nicotinamide (NAM) (two common NAD precursors) [23,24,25], NR has been found to induce no adverse effects in both human and animal objects [26,27,28]. NR supplementation promoted NAD metabolism and transcription of mitochondrial genes in the skeletal muscle while reduced the levels of circulating inflammatory cytokines in aged individuals [27]. NR can protect mitochondrial homeostasis through increasing the enzymatic activity of Sirtuin 1 (Sirt1) and Sirtuin 3 (Sirt3) in skeletal muscle and brown adipose tissue, and has a protective effect against the development of metabolic diseases caused by a high-fat diet (HFD) [29]. NR supplementation increases hepatocellular NAD levels and activates AMPK, which in turn enhances fatty acid oxidation (FAO) to inhibit ALD development [30]. Additionally, NR supplementation enhances the function of endothelial precursor cells to promote diabetic wound healing through activating AMPK [31]. However, the quantity of studies investigating the role of NR and AMPK in preventing IR is rather limited.

In this study, we aimed to identify whether NR can prevent IR and explore the potential molecular mechanisms involved. We hypothesized that the protective effect of NR against IR may be via inhibiting oxidative stress and improving mitochondrial function through the activation of the AMPK signaling pathway in the skeletal muscle.

## 2. Results

### 2.1. NR Supplementation Prevents Obesity-Related Metabolic Abnormalities in HFD-Fed Mice

To explore whether NR supplementation can protect against HFD-induced IR and obesity-related metabolic abnormalities in mice, we used NR (400 mg/kg·bw) treatment in HFD-fed mice for 24 weeks. The dosage of NR was based on our previous studies [32,33]. Mice fed with an HFD gained more body weight and fat mass than mice fed with the control diet (Figure 1A–C), whereas NR supplementation could alleviate HFD-induced increment of body weight and fat mass (Figure 1A–C). HFD increased serum lipid contents including total cholesterol (TC) and low-density-lipoprotein cholesterol (LDL-C) in mice, while NR supplementation inhibited this increase (Figure 1D,E). Similarly, NR supplementation alleviated liver TC and triglyceride (TG) accumulation induced by an HFD (Figure 1F,G). HFD also caused hepatic steatosis and enlarged the size of white adipocytes and brown adipocytes, whereas NR supplementation protected mice from these damages (Figure 1H).

### 2.2. NR Supplementation Alleviates Insulin Resistance in HFD-Fed Mice

HFD significantly increased the levels of fasting blood glucose (FBG) and fasting insulin in mice (Figure 2A,B). The HOMA-IR index of HFD-fed mice is significantly higher than mice in the control group, which indicates that HFD induced insulin resistance in mice (Figure 2C). Consistent with the results of the levels of circulation insulin (Figure 2B), the results of immunofluorescence in pancreas showed that HFD induced compensatory hyperfunction of insulin secretion in pancreatic β cells (Figure 2D). However, NR supplementation could decrease the FBG levels and the levels of fasting insulin both in circulation and in the pancreas. NR could prevent HFD-induced IR, which can be shown according to the HOMA-IR index (Figure 2A–D). To further explore the effect of NR on glucose tolerance and insulin sensitivity, IPGTT and ITT were performed. The results of IPGTT showed that HFD significantly impaired glucose tolerance in mice, whereas NR could protect mice against impaired glucose tolerance (Figure 2E). Additionally, HFD remarkably decreased insulin sensitivity in mice, while the improvement of insulin sensitivity due to NR compared with the HFD-fed group did not reach statistical significance (Figure 2F).

### 2.3. NR Supplementation Improves Insulin Sensitivity in Skeletal Muscle Cells

To further understand whether NR supplementation can improve insulin sensitivity in skeletal muscle cells, C2C12 myoblast cells were differentiated to myotube cells and then exposed to 0.25 mM PA for 24 h. Meanwhile, 0.5 mM NR was added to the medium together with PA. The dose of PA was selected through dose–response experiments (Appendix A). The dosage of NR was based on our previous studies [32,33]. The results of 2-NBDG uptake indicated that insulin could not stimulate PA-treated C2C12 myotube cells to uptake glucose, while NR improved the insulin sensitivity of C2C12 myotube cells and increased insulin-stimulated glucose uptake (Figure 3A). Meanwhile, PA-treated C2C12 myotube cells showed lower GLUT4 mRNA expression, but NR supplementation prevented this decrease (Figure 3B). Western blot results showed that PA treatment blocked the insulin signaling pathway and inhibited the activation of phosphorylated IRS1 and AKT after insulin stimulation. In contrast, NR supplementation increased the insulin-stimulated phosphorylation of AKT and promoted insulin signaling in C2C12 myotube cells (Figure 3C,D). Similarly, the phosphorylation of AKT was decreased in the skeletal muscle tissue of HFD-fed mice, while NR supplementation prevented this decrease (Figure 3E). Taken together, these results revealed that NR supplementation can improve insulin sensitivity in skeletal muscle cells.

### 2.4. NR Supplementation Alleviates Mitochondrial Dysfunction Induced by HFD

HFD-induced mitochondrial dysfunction has been recognized as one of the important mechanisms in the development of IR. Improving mitochondrial function might be effective in preventing IR. In our previous study, we have found that NR supplementation could improve mitochondrial function in the liver [32]. In this study, we found that NR supplementation can improve mitochondrial function in the skeletal muscle of mice with IR, which can be detected via mitochondrial function markers. Through the immunochemistry staining of TOM20 (a mitochondrial outer membrane protein), we found that HFD reduced the mitochondria amount in the skeletal muscle of mice, which was prevented via NR supplementation (Figure 4A). 4-Hydroxynonenal (4-HNE) is a biomarker of cellular oxidative stress. HFD induced oxidative stress in the skeletal muscle; this could be shown via the increased immunochemistry staining of 4-HNE, which was decreased after NR supplementation (Figure 4B). Moreover, we detected the protein expressions of key mediators involved in mitochondrial function including AMPK, PGC-1α and Sirt1. We found that the phosphorylation of AMPK decreased by nearly 50% and the protein expression of Sirt1 and PGC-1α in skeletal muscle was remarkably decreased in HFD-fed mice, whilst NR supplementation significantly increased the phosphorylation of AMPK and the protein expressions of Sirt1 and PGC-1α (Figure 4C,D).

### 2.5. NR Supplementation Ameliorates Mitochondrial Dysfunction and Oxidative Stress in Skeletal Muscle Cells through Activating AMPK

Consistent with the results of the animal experiment (Figure 4A), PA-treated C2C12 myotube cells had fewer active mitochondria than the control group, while NR supplementation could increase the number of active mitochondria in C2C12 myotube cells (Figure 5A). Further, we investigated the changes in mitochondrial membrane potential of C2C12 myotube cells. Through using the membrane-permeant JC-1 dye, we observed that fluorescence shifted from red to green as mitochondria became depolarized in PA-treated cells. NR supplementation resulted in an increased intensity ratio of red/green fluorescence, which indicated that NR increased the mitochondrial membrane potential and prevented mitochondrial depolarization in PA-treated C2C12 myotube cells (Figure 5B). Moreover, PA treatment remarkably increased mitochondrial ROS (mt-ROS) and malondialdehyde (MDA, a lipid peroxidation product) levels in C2C12 myotube cells (Figure 5C,D). NR supplementation significantly decreased the levels of mt-ROS and MDA, suggesting that NR could prevent the oxidative stress caused by PA treatment (Figure 5C,D).

To investigate the molecular mechanism of the improvement of mitochondrial function in NR-treated C2C12 myotube cells, NAD levels as well as protein and gene expression of key mediators involved in mitochondrial function were detected. We found that NAD levels decreased by nearly 40% after PA treatment, whereas NR supplementation could significantly increase NAD levels in the PA-treated C2C12 myotube cells (Figure 6A). AMPK is a key mediator of mitochondrial function, which is also important to the regulation of energy metabolism and IR. Consistent with the results of skeletal muscle tissue (Figure 4C,D), Western blot results revealed that PA treatment markedly decreased the phosphorylation of AMPK as well as the expression of the downstream protein of AMPK, including Sirt1 and PGC-1α (Figure 6B,C). NR supplementation prevented the above decrease. We also found that the mitochondrial biogenesis-related mRNA expression, including Sirt1, PGC-1α and TFAM, as well as the antioxidant-related mRNA expression, including SOD2 and Nrf2, were increased via NR supplementation compared with the PA-treated group (Figure 6D). These results indicated that AMPK activation may play an essential role in the improvement of mitochondrial function using NR.

### 2.6. AMPK Activation Is Required for Improvement on Mitochondrial Function and Insulin Sensitivity via NR in PA-Treated Skeletal Muscle Cells

To verify whether AMPK activation is required for NR to ameliorate mitochondrial dysfunction and IR in skeletal muscle cells, AMPK activity was inhibited using Compound C in C2C12 myotube cells prior to exposing them to PA and NR. We found that Compound C decreased the activity of AMPK by 50% (Figure 7A). The results of Western blotting showed that the effect of NR against PA-induced decreased protein expression of Sirt1 and PGC-1α was abrogated through inhibiting AMPK activation (Figure 7A). Compared with the PA-treated group, NR supplementation could significantly increase mitochondrial membrane potential in C2C12 myotube cells (Figure 7B). However, NR did not improve mitochondrial membrane potential after AMPK activity was inhibited (Figure 7B). Similarly, NR supplementation decreased PA-induced increment of the levels of mt-ROS, while this effect was not found after inhibiting AMPK activation (Figure 7C). These results suggest that AMPK activation is required for the improvement of mitochondrial function using NR in PA-treated C2C12 myotube cells.

PA treatment significantly decreased insulin sensitivity in C2C12 myotube cells. NR could markedly increase insulin-stimulated glucose uptake in PA-treated C2C12 myotube cells, while this protective effect was abolished after the inhibition of AMPK activity (Figure 8A). Moreover, compared with the PA-treated group, NR could remarkably increase the phosphorylation of AKT and promoted insulin signaling transduction. Compound C pre-treatment inhibited the effect of NR on promoting insulin signaling transduction, revealed in the form of reduced phosphorylation of AKT after insulin stimulation compared to the PA + NR group (Figure 8B,C). The above results indicate that AMPK activation is required for the improvement of insulin sensitivity using NR in PA-treated C2C12 myotube cells.

## 3. Discussion

Due to the growing prevalence of obesity and IR, more and more people suffer from T2DM. As such, an effective treatment for obesity and IR is of great urgency. In this study, we found that NR could effectively prevent IR in both in vivo and in vitro experiments. NR replenished skeletal muscle cellular NAD pool, boosted the activities of AMPK, Sirt1 and PGC-1α, reduced oxidative stress, restored mitochondrial function and finally alleviated IR.

NAD is considered a key signaling molecule as well as a rate-limiting substrate for a variety of enzymes involved in various biological processes. Inadequate NAD biosynthesis in skeletal muscle, adipose tissue and the liver leads to the pathogenesis of obesity-associated metabolic abnormalities, including IR [18,34]. Replenishing NAD might be an effective way to prevent IR. NR, the key NAD intermediate, has been found to improve glucose tolerance and insulin sensitivity as well as to prevent NAFLD in mice [35]. NR administration could also alleviate IR, reduce cardiometabolic risk factors and modulate cardiac oxidative stress in obese rats [36]. These studies are consistent with our data that treatment with NR can improve insulin sensitivity in rodents. Additionally, promoting NAD biosynthesis through administering nicotinamide mononucleotide (NMN) for 10 weeks increased insulin signaling and sensitivity (increased insulin-stimulated glucose disposal rate) in the skeletal muscle of obese/overweight prediabetic women [20]. These previous studies found that replenishing NAD using NR or NMN increased insulin sensitivity in rodents or humans. However, the above studies did not further explore the potential mechanisms of the amelioration of IR after applying NR or other NAD intermediates. Consistent with these studies, our results demonstrated that NR supplementation for 24 weeks effectively decreased the levels of FBG and fasting insulin, improved glucose tolerance and increased muscle insulin sensitivity (increased phosphorylated AKT) in HFD-fed mice. NR supplementation for 24 h significantly increased insulin signaling (increased expression of insulin-stimulated phosphorylated AKT) and insulin-simulated glucose uptake in PA-treated C2C12 myotube cells. Our results further illustrated that the potential mechanism of the improvement of NR on IR could be related to NR effectively activating AMPK in skeletal muscle cells, which in turn improved mitochondrial dysfunction and oxidative stress to improve IR.

AMPK is involved in glucose uptake and is critical for maintaining mitochondrial function in skeletal muscle. Skeletal muscle-specific AMPK knockout mice showed decreased mitochondrial content, impaired mitochondrial capacity and contraction-stimulated glucose uptake [37]. Using MK-8722 to activate AMPK in skeletal muscle could increase glucose uptake and glycogen synthesis [38]. Irisin, a myokine, could activate AMPK and extracellular signal-regulated kinase ½ (Erk1/2) in skeletal muscle cells to increase glucose uptake and glycogen accumulation in response to insulin stimulation [39]. Our data also demonstrated that NR supplementation could increase insulin-stimulated glucose uptake in skeletal muscle cells in an AMPK-dependent manner. Moreover, our results found that NR supplementation can prevent the decreased mRNA expression of GLUT4 induced by PA, which may further promote insulin-stimulated glucose uptake in skeletal muscle cells. However, whether the effect of NR on increasing the mRNA expression of GLUT4 in skeletal muscle cells depends on the activation of AMPK needs further experiments. Additionally, activation of AMPK enhanced the transcription of PGC-1α and the activity of its coactivators, and increased mitochondrial biogenesis, thereby promoting muscle regeneration and improved energy metabolism to inhibit apoptosis in skeletal muscle cells [40,41]. Our study found that in both HFD-fed mice and PA-treated C2C12 myotube cells, the phosphorylation of AMPK decreased by nearly 50%, which led to mitochondrial dysfunction and IR. NR supplementation significantly activated AMPK in skeletal muscle cells, which in turn activated the downstream Sirt1 and PGC-1α to improve mitochondrial function and alleviate IR. Our data indicated that AMPK activation may play a significant role in the development of mitochondrial dysfunction and IR.

Many studies have demonstrated the abnormalities of mitochondrial function and/or content in the skeletal muscle of humans with IR or T2DM [3,6,42,43]. Saturated fat uptake and/or excess fat exposure led to incomplete FAO in mitochondria and preferentially generated “toxic” diacylglycerol (DAG), which disturbs the insulin signaling and promotes IR in skeletal muscle [44]. Lipids can activate skeletal muscle mitochondrial fission and quality control networks to induce IR in humans [45]. Reduced mitochondrial biogenesis contributes to obvious alterations in the numbers and functions of mitochondria. Meanwhile, decreased expression of PGC-1α/β and compromised mitochondrial biogenesis could lead to decreased levels of oxidative phosphorylation (OXPHOS), which affected the energy metabolism and insulin signaling in skeletal muscle [46,47]. Resveratrol could activate the Sirt1/PGC-1α pathway, which stimulates mitochondrial biogenesis to improve IR in skeletal muscle [48]. Considering that NAD biology is important for mitochondrial function, and the strong association between the function of mitochondria and insulin sensitivity in muscle [49], we examined the effect of NR on skeletal muscle mitochondrial function. The results confirmed that NR could prevent a decrease in the amount of mitochondria in HFD-fed mice, and could protect mitochondria from damage induced by PA treatment in C2C12 myotube cells. Moreover, NR supplementation could activate AMPK to increase PGC-1α protein expression and mitochondrial biogenesis-related mRNA (Sirt1, PGC-1α, TFAM) expression. In general, our data indicated that NR can improve mitochondrial function in skeletal muscle, probably via AMPK activation.

Mitochondria are one of the major sites of ROS production in cells [50]. During chronic nutrient oversupply, the amount of nutrients ingested far exceeds the demand of ATP and the uncoupling capacity, which induces incomplete FAO and increases mitochondrial ROS production [8,51], further contributing to compromised mitochondrial activity and blocking downstream insulin signaling in T2DM individuals [43,52,53,54]. Additionally, increased ROS has been largely reported to be related to IR in animal models. HFD feeding has been reported to increase mitochondrial H_2_O_2_ production and decrease redox-buffering capacity, oxidative capacity and ATP levels in the skeletal muscle of mice [55,56,57,58]. Nonetheless, blocking mt-ROS production or scavenging ROS has been found to be able to prevent IR [59,60]. Superoxide dismutase 2 (SOD2) is known as an important cellular defense enzyme against oxidative stress. Nuclear respiratory factor 2 (Nrf2) is recognized as one of the significant redox regulators which can regulate its downstream target genes, including heme oxygenase-1 (HO-1) and NAD(P)H quinone dehydrogenase 1 (NQO1), to mediate ROS production [61]. Our results showed that NR facilitated mitochondrial antioxidant defenses to prevent HFD/PA-induced ROS production through increasing SOD2 and Nrf2 gene expression. Previous studies suggested that PGC-1α can regulate hundreds of genes that combat oxidative stress as well as enzymes such as SOD2 and catalase, which can directly detoxify ROS [62]. In our study, NR activated AMPK and increased PGC-1α expression. Additionally, both AMPK and PGC-1α can regulate the expression of Nrf2 [62,63]. As such, the AMPK/PGC-1α/SOD2 and AMPK/PGC-1α/Nrf2 signaling pathway could be key ways to maintain mt-ROS homeostasis in NR-treated skeletal muscle cells. Furthermore, AMPK is of great importance for clearing ROS because it can activate mitophagy to degrade the damaged mitochondria to scavenge ROS and inhibit oxidative stress and apoptosis [64]. Whether NR can activate AMPK and then regulate mitophagy to maintain mt-ROS homeostasis in skeletal muscle cells is to be investigated in our next-step research.

Our results revealed that NR supplementation can replenish NAD to prevent IR in HFD-fed mice, which can be further validated in human trials on obese/overweight individuals or aged individuals whose NAD levels might be lower than those of healthy people. Additionally, our study figured out that the potential mechanism of the amelioration of IR using NR is NR-mediated AMPK activation in the skeletal muscle. The essential role of AMPK activation in the preventive effects of NR on IR can be explored in skeletal-muscle-specific AMPK knockout mice in the future.

In conclusion, our research proposed a potential mechanism of the amelioration of IR using NR-mediated AMPK activation in skeletal muscle. NR can activate the AMPK pathway, restore mitochondrial function, alleviate oxidative stress and reverse IR induced by HFD.

## 4. Materials and Methods

### 4.1. Animal Experiments

Experimental protocols were reviewed and approved by the Sun Yat-sen University Animal Ethics Committee. Male C57BL/6J mice were purchased from the Guangdong experiment animal center (Guangzhou, China). Mice were housed in a room with a 12–12 h light–dark cycle and temperature of 25 ± 2 °C. C57BL/6 male mice (8 weeks old) were randomly allocated into three groups: control (13% kcal fat), HFD (60% kcal fat), and HFD + NR (*n* = 5–7). NR (ChromaDex, Inc., Los Angeles, CA, USA) was administered via gavage at 400 mg/kg/d B·W for 24 weeks. The dosage of NR was based on our previous studies [32,33]. At the end of the experiment, all mice were fasted overnight prior to sacrifice. Livers, gastrocnemii and plasma were harvested and stored at −80 °C for further investigation. Liver, gastrocnemius, brown adipose tissue and white adipose tissue samples were fixed in formalin for H&E staining or IHC.

#### 4.1.1. Intraperitoneal Glucose Tolerance Test (IPGTT) and Insulin Tolerance Test (ITT)

After overnight fasting, mice were given an intraperitoneal injection of glucose (2 g/kg B·W) for IPGTT. ITTs were performed seven days after the IPGTT assessment. Following a 6 h fast, mice were given an intraperitoneal injection of human insulin (1 U/kg B·W). The levels of fasting blood glucose (0 min) were detected before the glucose/insulin injection. Then, the levels of blood glucose of mice were quantified at 15, 30, 60, 90 and 120 min after the administration of insulin or glucose.

#### 4.1.2. Immunohistochemistry

The gastrocnemius muscle sections were deparaffinized in xylene and hydrated in ethanol. After washing with PBS, the sections were incubated in 5% goat serum for 30 min at 37 °C and then treated with antibodies against TOM20 (1:100, CST, Beverly, MA, USA) and 4-Hydroxynonenal (4-HNE) (1:100, CST) overnight at 4 °C. The sections were treated with the secondary antibody IgG-HRP (1:500, CST) for 30 min at 37 °C and washed with PBS. Immunostaining was performed using 3,3′ diaminobenzidine tetrahydrochloride (DAB).

#### 4.1.3. Immunofluorescence

The pancreas sections were deparaffinized in xylene and hydrated in ethanol. After washing with PBS, the sections were incubated in 5% goat serum for 30 min at 37 °C and then treated with antibodies against insulin (1:200, Abcam) and glucagon (1:200, CST) overnight at 4 °C. The sections were treated with the secondary antibodies Alexa Fluor^®^ 488 and Alexa Fluor^®^ 594 (1:500, Thermo Fisher Scientific, Waltham, MA, USA) for 30 min at 37 °C in a dark room and then washed with PBS. DAPI was used to label the nucleus. The images were photographed with a confocal microscope (Leica TCS SP5).

### 4.2. Cell Culture Experiments

Mouse skeletal muscle C2C12 myoblast cells were purchased from ATCC and cultured in Dulbecco’s modified Eagle medium (DMEM; Gibco-Invitrogen, Carlsbad, CA, USA) containing 10% fetal bovine serum (FBS) and 1% penicillin-streptomycin in 5% CO_2_ at 37 °C. NR were obtained from ChromaDex, Inc. (Los Angeles, CA, USA). Palmitic acid was obtained from Sigma Chemical (St. Louis, MO, USA). For C2C12 cells differentiation: the medium was replaced with DMEM containing 2% horse serum for 4~5 days. When the cells differentiated completely, they were exposed to 0.25 mM PA with or without 0.5 mM NR for 24 h. The dose of PA was selected through dose–response experiments. The dosage of NR was based on our previous studies [32,33]. To explore the role of AMPK, cells were pretreated with 10µM Compound C (Selleck, Houston, TX, USA) for 8 h before PA and NR treatment. The dosage of Compound C was from the one commonly used in the references.

#### 4.2.1. Insulin-Stimulated Glucose Uptake Analysis

Cells were starved in KRBH buffer without glucose for 3 h and then treated with 100 nM insulin for 30 min. The cells were washed with PBS three times and then treated with 100 μM 2-NBDG for 30 min. Cells were washed three times. The images were photographed with a Leica DMI8 fluorescence microscope.

#### 4.2.2. MitoTracker Red Staining

Cells were incubated with DMEM medium with 400 nM MitoTracker Red (Thermo Fisher Scientific, Waltham, MA, USA) at 37 °C for 45 min. Then, the cells were fixed in ice-cold methanol for 15 min, washed with PBS three times and then incubated in 600 nM DAPI for 5 min in a darked room. The images were photographed with a Leica DMI8 fluorescence microscope.

#### 4.2.3. Mitochondrial Membrane Potential Determination

The mitochondrial membrane potential was measured using Mitochondrial Membrane Potential Detection Kit JC-1(Beyotime, Shanghai, China) following the manufacturer protocol. Cells were photographed using a Leica DMI8 Fluorescence Microscope. The mitochondrial membrane potential was represented as the ratio of red to green fluorescence intensity.

#### 4.2.4. MitoSox Staining

Cells were incubated with DMEM medium with 5 μM MitoSOX Red and 2 mg/L Hoechst (Thermo scientific, Waltham, MA, USA) for 10 min at 37 °C. Then, the levels of mitochondrial ROS were evaluated using a Leica DMI8 fluorescence microscope.

#### 4.2.5. Measurement of MDA Levels

Determination of malondialdehyde (MDA) level in C2C12 myotube cells was performed using a lipid peroxidation MDA assay kit (Beyotime, Shanghai, China). Briefly, 100 μL extract of C2C12 myotube cells was mixed with 200 μL of MDA working solution, incubated in a 100 °C for 40 min and then cooled with running water. The mixture was centrifuged at 1078 g for 10 min and the supernatant was used to measure absorbance at 532 nm.

#### 4.2.6. Measurement of NAD Levels

Determination of NAD levels in C2C12 myotube cells was performed using a NAD assay kit (Beyotime, Shanghai, China) following the manufacturer protocol. The NAD levels were quantified at 450 nm absorbance.

#### 4.2.7. Western Blot Analysis

Total proteins were extracted via RIPA containing protease inhibitor PMSF (Beyotime, Shanghai, China) and PhosSTOP phosphatase inhibitor (Beyotime, Shanghai, China). The protein concentration was measured using a BCA assay kit (Beyotime, Shanghai, China). Antibodies against p-AMPKα (Th172), AMPKα, Sirt1, p-AKT, AKT and IRS1 were from cell signaling technology (CST, Beverly, MA, USA). Antibody against p-IRS1 was from Thermo scientific (Waltham, MA, USA). Protein signals were visualized using an enhanced chemiluminescence detection system according to the manufacturer’s instructions (ECL, Thermo Fisher Scientific, Waltham, MA, USA)

#### 4.2.8. Quantitative PCR

The total RNA of skeletal muscle tissue and C2C12 myotube cells were extracted using Trizol reagent (Thermo Fisher Scientific, Waltham, MA, USA). Prime Script RT master mix (Takara, Dalian, China) was used for the reverse transcription of RNA to cDNA. qPCR was performed with SYBR Premix Ex Taq II (Takara, Dalian, China). All of the samples had three parallel wells, and the experiments were repeated at least three times. β-actin was employed as the internal standard to normalize gene expression via the 2−∆∆Ct method. Primers for qPCR analysis were designed using the NCBI database. A complete list of mice primers is given in Table 1.

### 4.3. Statistical Analysis

Data analysis was performed with SPSS 25.0 software (Chicago, IL, USA). All experimental data are expressed as mean ± SEM. Results were analyzed using one-way ANOVA followed by Bonferroni’s post hoc multiple-comparisons test to determine statistical significance. Comparison between two experimental groups was based on the two-tailed *t* test. *p* < 0.05 was considered statistically significant. Each experiment was repeated at least three times.

## 5. Conclusions

Mitochondrial dysfunction and oxidative stress in skeletal muscle are the key mechanisms of IR. NAD is of great importance for mitochondrial function, redox homeostasis and insulin sensitivity in skeletal muscle. NR is an effective NAD booster that can be easily obtained in daily life through food or dietary supplements. Our research found that NR supplementation replenished NAD potentially via activation of the AMPK pathway to prevent IR in HFD-fed mice and PA-treated C2C12 myotube cells. Supplementing NAD boosters such as NR, which can effectively increase NAD levels, restore mitochondrial function and alleviate oxidative stress in skeletal muscle, might be a potential preventive strategy for IR.

## Figures and Tables

**Figure 1 ijms-24-10015-f001:**
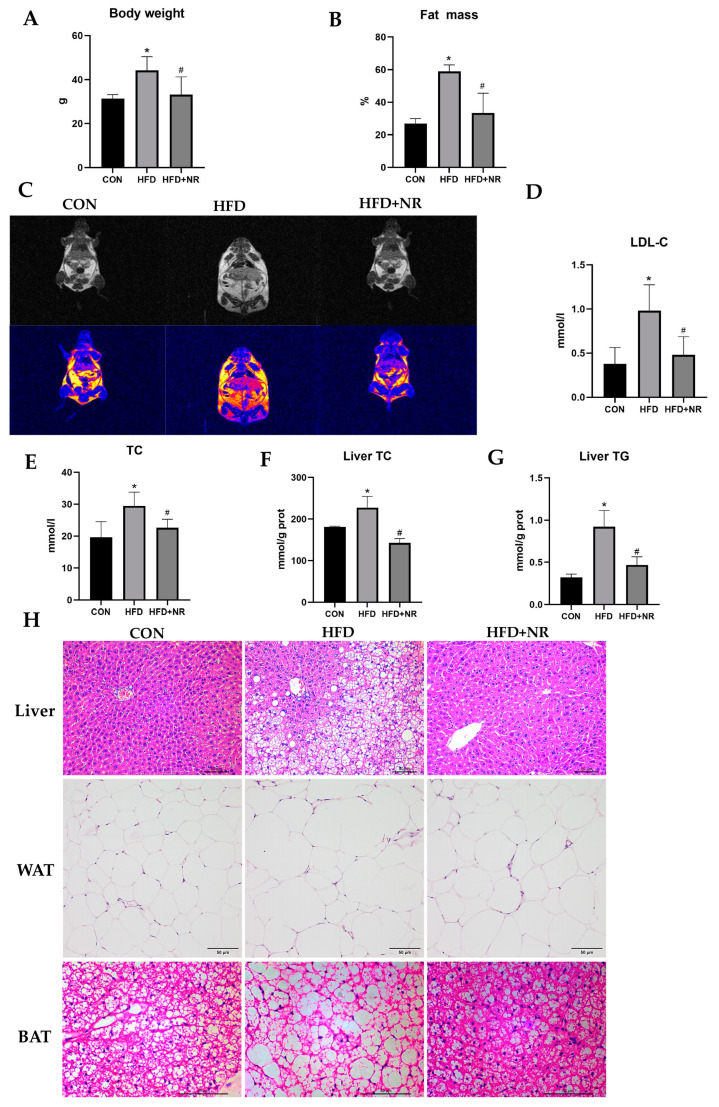
NR supplementation prevents obesity-related metabolic abnormalities in HFD-fed mice. Mice were fed with an HFD ± NR supplementation for 24 weeks. (**A**) Body weight in the indicated groups. (**B**) Fat body mass was determined via visceral adiposity index. (**C**) Body composition imaging of mice, the yellow part is the fat. (**D**) Serum LDL-C contents. (**E**) Serum TC contents. (**F**) Liver TC contents. (**G**) Liver TG contents. (**H**) H&E staining of liver, WAT, BAT. For the H&E staining of liver and WAT, the microscope magnification is 200×. For the H&E staining of WAT and BAT, the microscope magnification is 400×. *n* = 5–7 in each group. * *p* < 0.05 compared with the CON group; ^#^ *p* < 0.05 comparing HFD group vs. HFD + NR group. Abbreviations: CON—control group; HFD—high-fat diet group, HFD + NR—high-fat diet + nicotinamide riboside group; LDL-C—low-density-lipoprotein cholesterol; TC—total cholesterol; TG—triglyceride; H&E staining—hematoxylin–eosin staining; WAT—white adipose tissue; BAT—brown adipose tissue.

**Figure 2 ijms-24-10015-f002:**
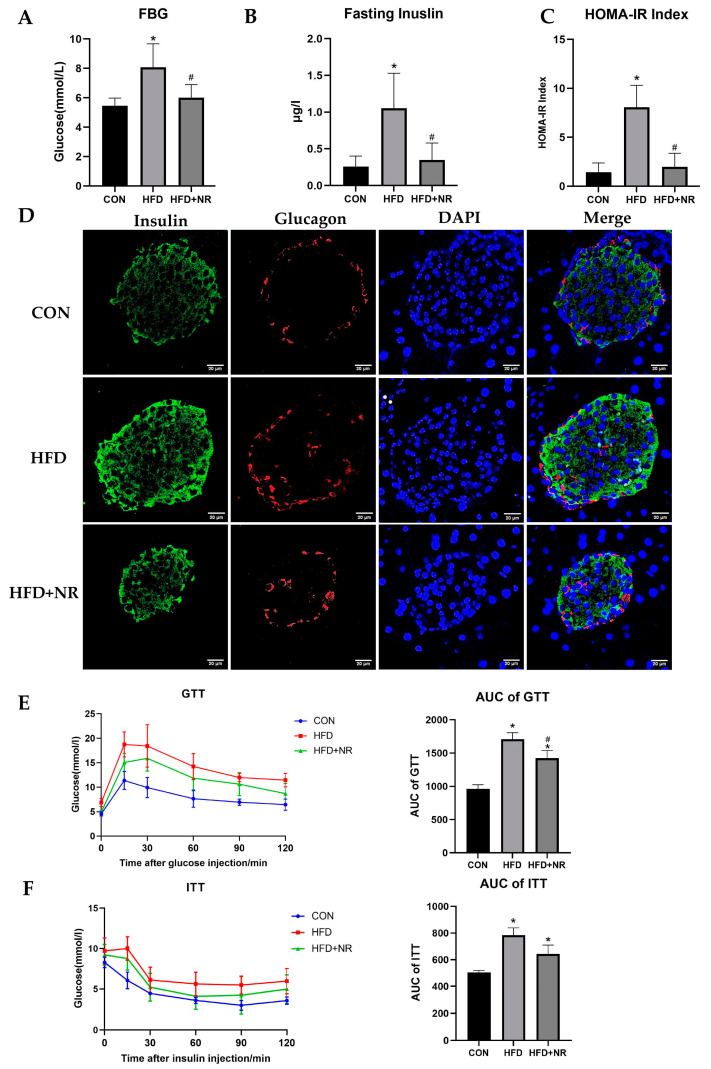
NR supplementation alleviates insulin resistance in HFD-fed mice. (**A**) Fasting blood glucose levels. (**B**) Fasting blood insulin levels. (**C**) HOMA-IR index. (**D**) Immunofluorescence of insulin and glucagon in the islet of mice. Images were taken using a confocal microscope, the microscope magnification is 400×. (**E**,**F**) Representative IPGTT (**E**) and ITT (**F**) and the AUC result of mice in the indicated groups. *n* = 5–7 for each group. * *p* < 0.05 compared with the CON group; ^#^ *p* < 0.05 comparing HFD group vs. HFD + NR group. Abbreviations: FBG—fasting blood glucose levels; HOMA-IR index—homeostasis model assessment of insulin resistance index; IPGTT—intraperitoneal glucose tolerance test; ITT—insulin tolerance test; AUC—area under the curve.

**Figure 3 ijms-24-10015-f003:**
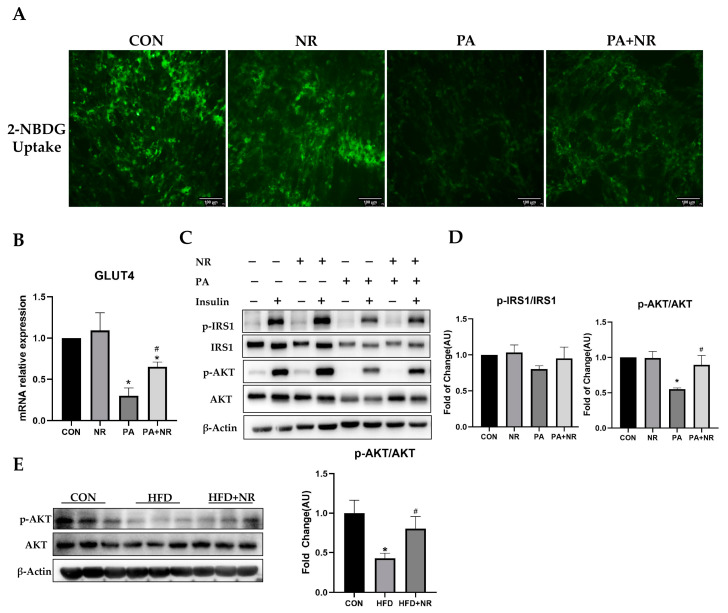
NR supplementation improves insulin sensitivity in skeletal muscle cells. (**A**) Uptake of 2-NBDG by C2C12 myotube cells, the microscope magnification is 200×. (**B**) Relative mRNA expression of GLUT4 gene. (**C**) Representative Western blot results of insulin signaling pathway–associated protein (IRS1, AKT) of cells. β-Actin served as loading control. (**D**) Quantitative results for WB; *n* = 3 in each group. (**E**) Representative Western blot results and quantitative results for WB of AKT of skeletal muscle tissue of mice. β-Actin served as a loading control; *n* = 3 in each group. * *p* < 0.05 compared with the CON group; ^#^ *p* < 0.05 comparing PA group vs. PA + NR group. Abbreviations: 2-NBDG—2-deoxy-2-[(7-nitro-2,1,3-benzoxadiazol-4-yl)-amino]-D-glucose; GLUT4—glucose transporter type 4; IRS1—insulin receptor substrate 1; p-IRS1—phosphorated insulin receptor substrate 1; AKT—protein kinase B; p-AKT—phosphorated protein kinase B.

**Figure 4 ijms-24-10015-f004:**
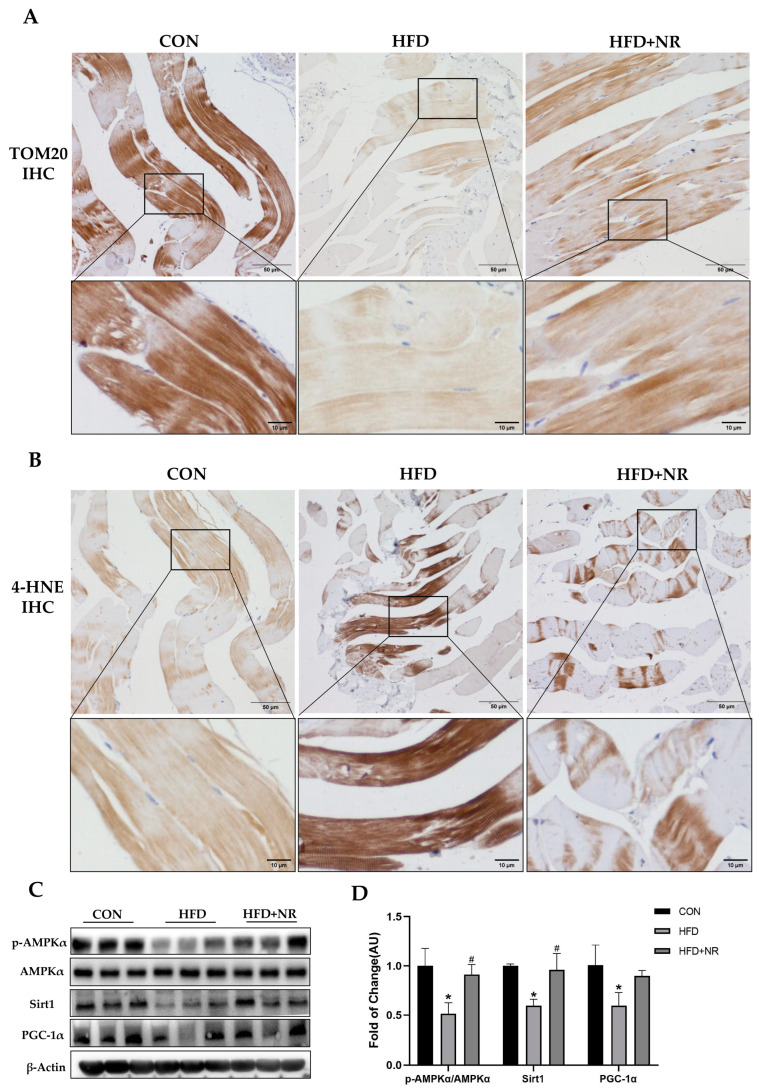
NR supplementation alleviates mitochondrial dysfunction induced by HFD. (**A**) IHC staining of TOM20 in skeletal muscle tissue, the microscope magnification is 200×. (**B**) IHC staining of 4-HNE in skeletal muscle tissue, the microscope magnification is 200×. (**C**) Representative Western blot results of p-AMPKα, AMPKα, Sirt1 and PGC-1α of skeletal muscle tissue of mice. β-Actin served as a loading control. (**D**) Quantitative results for WB. *n* = 3 for each group. * *p* < 0.05 compared with the CON group; ^#^ *p* < 0.05 comparing HFD group vs. HFD + NR group. Abbreviations: IHC—immunohistochemistry; TOM20—translocase of the outer mitochondrial membrane complex 20; 4-HNE—4-Hydroxynonenal; AMPKα—adenosine monophosphate-activated protein kinase alpha; p-AMPKα—phosphorated adenosine monophosphate-activated protein kinase alpha; Sirt1—sirtuin 1; PGC-1α—peroxisome proliferator-activated receptor gamma coactivator 1-alpha.

**Figure 5 ijms-24-10015-f005:**
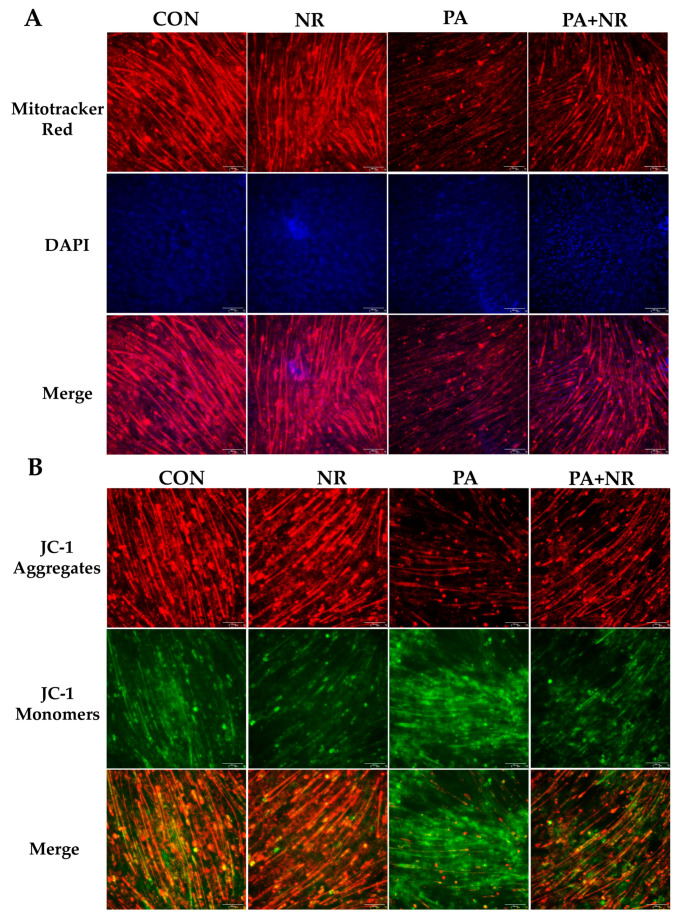
NR supplementation ameliorates mitochondrial dysfunction and oxidative stress in muscle cells. (**A**) Mitotracker Red staining, the microscope magnification is 200×. (**B**) JC-1 staining, the microscope magnification is 200×. (**C**) MitoSOX staining, the microscope magnification is 200×. (**D**) MDA levels. n = 3 for each group. * *p* < 0.05 compared with the CON group; ^#^ *p* < 0.05 comparing PA group vs. PA + NR group. Abbreviations: DAPI—4′,6-diamidino-2-phenylindole; MDA—malondialdehyde; JC-1—5,5′,6,6′-Tetrachloro-1,1′,3,3′-tetraethyl-imidacarbocyanine iodide; MitoSOX—mitochondrial superoxide indicators.

**Figure 6 ijms-24-10015-f006:**
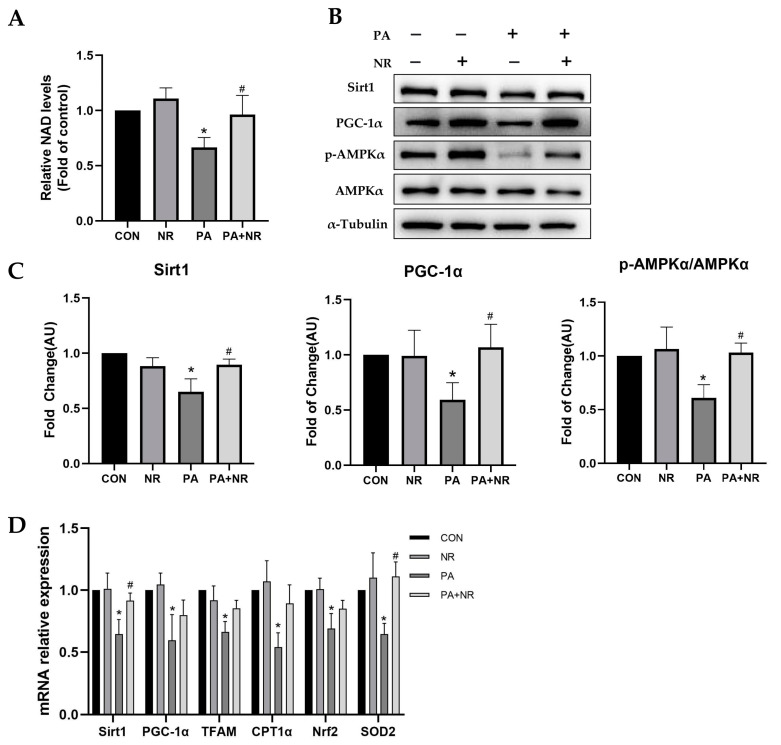
NR supplementation ameliorates mitochondrial dysfunction and oxidative stress in skeletal muscle cells through activating AMPK. (**A**) Relative NAD levels. (**B**) Representative Western blot results of Sirt1, p-AMPKα, AMPKα and PGC-1α of C2C12 myotube cells. α-Tubulin served as a loading control. (**C**) The quantitative results for WB. (**D**) Relative mRNA expression of Sirt1, PGC-1α, TFAM, Nrf2, SOD2 and CPT1α in C2C12 myotube cells. n = 3 for each group. * *p* < 0.05 compared with the CON group; ^#^ *p* < 0.05 comparing PA group vs. PA + NR group. Abbreviations: NAD—nicotinamide adenine dinucleotide; TFAM—mitochondrial transcription factor A; Nrf2—nuclear respiratory factor 2; SOD2—superoxide dismutase 2; CPT1α—carnitine palmitoyltransferase 1α.

**Figure 7 ijms-24-10015-f007:**
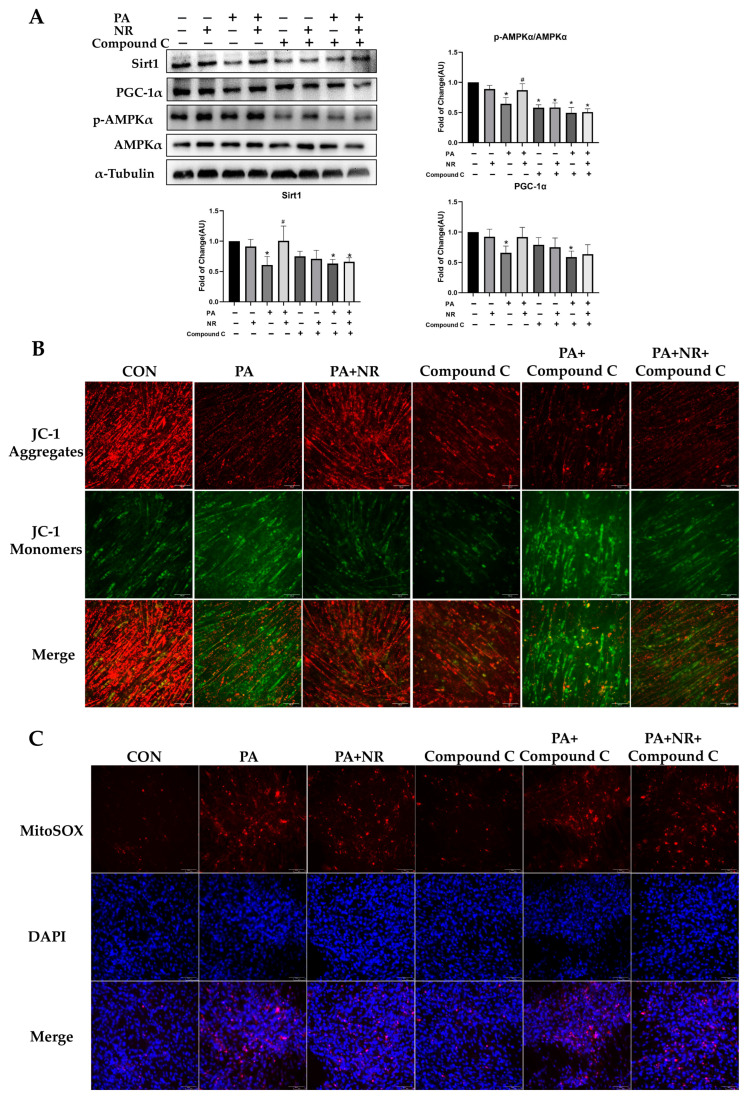
AMPK activation is required for improvement of mitochondrial function using NR in PA-treated skeletal muscle cells. (**A**) Representative Western blot results for Sirt1, p-AMPKα, AMPKα and PGC-1α in C2C12 myotube cells and the quantitative results for WB. α-Tubulin served as a loading control. (**B**) JC-1 staining, the microscope magnification is 200×. (**C**) MitoSOX staining, the microscope magnification is 200×. Compound C means that the C2C12 myotube cells were treated with 10μM Compound C for 8 h before adding PA and NR. * *p* < 0.05 compared with the CON group; ^#^ *p* < 0.05 comparing PA group vs. PA + NR group. Abbreviations: Compound C—6-[4-(2-Piperidin-1-ylethoxy) phenyl]-3-pyridin-4-ylpyrazolo [1,5-a] pyrimidine.

**Figure 8 ijms-24-10015-f008:**
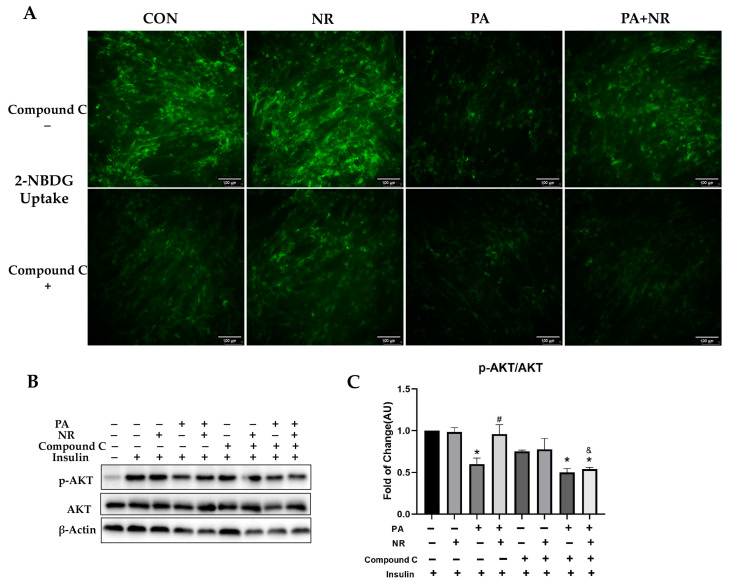
AMPK activation is required for improvement of insulin sensitivity using NR in PA-treated skeletal muscle cells. (**A**) 2-NBDG uptake of C2C12 myotube cells in the indicated groups, the microscope magnification is 200×. (**B**) Representative Western blot results of insulin signaling pathway-associated protein (AKT) of cells. β-Actin served as a loading control. (**C**) Quantitative results for WB. * *p* < 0.05 compared with the CON group; ^#^ *p* < 0.05 comparing PA group vs. PA + NR group. ^&^ *p* < 0.05 PA + NR group vs. PA + NR + Compound C group.

**Table 1 ijms-24-10015-t001:** Primers used for the RT-PCR analysis.

Genes	Primers	Sequences (Primer: 5′-3′)
SOD2	Forward primer	GCCCAAACCTATCGTGTCCA
	Reverse primer	AGGGAACCCTAAATGCTGCC
β-Actin	Forward primer	GTGGTGGTGAAGCTGTAGCC
	Reverse primer	AGCCATGTACGTAGCCATCC
Sirt1	Forward primer	TGTGAAGTTACTGCAGGAGTGTAAA
	Reverse primer	GCATAGATACCGTCTCTTGATCTGAA
PGC-1α	Forward primer	AAGTGTGGAACTCTCTGGAACTG
	Reverse primer	GGGTTATCTTGGTTGGCTTTATG
TFAM	Forward primer	AACACCCAGATGCAAAACTTTCA
	Reverse primer	GACTTGGAGTTAGCTGCTCTTT
Nrf2	Forward primer	CTTTAGTCAGCGACAGAAGGAC
	Reverse primer	AGGCATCTTGTTTGGGAATGTG
CPT-1α	Forward primer	CTCCGCCTGAGCCATGAAG
	Reverse primer	CACCAGTGATGATGCCATTCT
GLUT4	Forward primer	GTGACTGGAACACTGGTCCTA
	Reverse primer	CCAGCCACGTTGCATTGTAG

## Data Availability

The data presented in this study are available on request from the corresponding author.

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
