# Peer review of "Improving Mitochondrial Function in Skeletal Muscle Contributes to the Amelioration of Insulin Resistance by Nicotinamide Riboside"

_ijms, 2023, doi:10.3390/ijms241210015_

Round 1

Reviewer 1 Report

In this manuscript, Li and colleagues, demonstrated that the improvement of the mitochondrial function via AMPK pathway activation in skeletal muscle plays a crucial role in the amelioration of IR by nicotinamide riboside (NR). The manuscript is well organized and easy to read. The authors have used two experimental systems investigating in ex vivo and in an in vitro model, the role exercised by NR on restored of the skeletal muscle function. I have only some minor suggestions to improve the quality of manustript:

1) The authors could use the figure 9 like as a graphical abstract. The graphical abstract could allow readers to quickly gain an understanding of the take-home message of the paper.

2) The author could indicate the reference about the chosen of dose of NR injected in murine model.

    3) In figure 7A, the blot representative of the Tubulin seem to be inverted. I would want see the original blot. Furthermore, the authors should insert the type of tubulin used.

    4) In figure 8A, the blot representative of the β-Actin seem to be inverted. I would see the original blot.

Author Response

Author's Reply to the Review Report

Thanks for the suggestions of reviewers! We have revised our manuscript and the revised part were marked in red font on the manuscript. Here are our responses for the questions of the reviewer. The image of WB blot can‘t be inserted in this note, but reviewers and editors can download the word file to see the image.

In this manuscript, Li and colleagues, demonstrated that the improvement of the mitochondrial function via AMPK pathway activation in skeletal muscle plays a crucial role in the amelioration of IR by nicotinamide riboside (NR). The manuscript is well organized and easy to read. The authors have used two experimental systems investigating in ex vivo and in an in vitro model, the role exercised by NR on restored of the skeletal muscle function. I have only some minor suggestions to improve the quality of manuscript:

  1. The authors could use the figure 9 like as a graphical abstract. The graphical abstract could allow readers to quickly gain an understanding of the take-home message of the paper.

Reply: Thank you! As suggested, figure 9 has been submitted as a graphical abstract and we deleted the figure 9 on the manuscript.

  1. The author could indicate the reference about the chosen of dose of NR injected in murine model.

Reply: Thanks! The references about the chosen of dose of NR administration in murine model have been indicated in the 4.1. Animal Experiments section on the page 19.

  1. In figure 7A, the blot representative of the Tubulin seem to be inverted. I would want see the original blot. Furthermore, the authors should insert the type of tubulin used.

Reply: Thank you! In figure 7A, the blot representative of the Tubulin was not inverted. The type of tubulin used is α- Tubulin. The original blot is shown below.The image of WB blot can‘t be inserted in this note, but reviewers and editors can download the word file to see the image.

  1. In figure 8A, the blot representative of the β-Actin seem to be inverted. I would see the original blot.

Reply: Thanks! In figure 8A, the blot representative of the β-Actin was not inverted. The original blot is shown below.The image of WB blot can‘t be inserted in this note, but reviewers and editors can download the word file to see the image.

Reviewer 2 Report

After a thorough review of the manuscript, it is clear that the research presented is of significant value, contributing to our understanding of the potential mechanisms through which nicotinamide riboside (NR) might ameliorate insulin resistance (IR). However, there are several areas of the manuscript that require substantial revision for clarity, depth, and robustness. The following points outline specific areas that need addressing:

1. Methods - Animal Experiment Details: The manuscript needs more details about the experimental design for the animal study. Clarify how the animals were randomized and how potential confounding factors were controlled.

2. Methods - Sample Size Justification: The manuscript does not provide a justification for the sample size used in animal experiments. Consider providing a power analysis or other rationale for the chosen sample size.

3. Discussion - Lack of Comparison to Previous Studies: The discussion does not adequately compare the results to those of previous studies. Including such a comparison would help place the results in the context of the existing literature.

4. Discussion - Future Directions: The discussion lacks suggestions for future research. The authors should discuss the implications of their findings and propose potential directions for future studies.

5. Conclusions - Lack of Context: The conclusions do not provide a broader context for the findings. Discussing how the findings could influence the development of therapeutic strategies and future studies would strengthen the conclusion.

 Overall Structure and Language: The manuscript should be thoroughly proofread to ensure that it is free of typographical errors and that it maintains a coherent and logical flow. Addressing these points will greatly improve the manuscript's clarity, robustness, and impact. 

Minor editing of English language required

Author Response

Author's Reply to the Review Report

Thanks for the suggestions of reviewers! We have revised our manuscript and the revised part were marked in red font on the manuscript. Here are our responses for the questions of the reviewer.

After a thorough review of the manuscript, it is clear that the research presented is of significant value, contributing to our understanding of the potential mechanisms through which nicotinamide riboside (NR) might ameliorate insulin resistance (IR). However, there are several areas of the manuscript that require substantial revision for clarity, depth, and robustness. The following points outline specific areas that need addressing:

  1. Methods - Animal Experiment Details: The manuscript needs more details about the experimental design for the animal study. Clarify how the animals were randomized and how potential confounding factors were controlled.

Reply: Thank you! There were 18 male C57BL/6J mice (8 weeks old, weighted 22.3~26.6g) in our animal experiment. The mice were completely randomized to be assigned to each group. The randomization process is as the following:

First, the mice were weighed and sorted in order;

Numbering: number the mice in 1, 2, 3...6, and so on, numbering up to 18;

Using the random number table for grouping: These 18 mice were randomly assigned to 3 groups (CON group, HFD group, HFD+NR group). The grouping method used the random number table, and distributed to each group according to the remainder of the random number. The 15 mice were distributed to 3 groups. For the remaining 3 mice, considering that the high-fat diet may cause health issues, we distributed 2 mice to the HFD group and 1 mouse to the HFD+NR group. So there were 5 mice in the CON group, 7 mice in the HFD group, 6 mice in the HFD+NR group.

  1. Methods - Sample Size Justification: The manuscript does not provide a justification for the sample size used in animal experiments. Consider providing a power analysis or other rationale for the chosen sample size.

Reply: Thanks for this suggestion. We have done a preliminary experiment to explore the intervention methods of animal experiments and found that 5 mice in each group is enough to reach statistical differences. We also considered animal ethics and welfare, so we chose 5-7 mice in each group in the end.

  1. Discussion - Lack of Comparison to Previous Studies: The discussion does not adequately compare the results to those of previous studies. Including such a comparison would help place the results in the context of the existing literature.

Reply: Thanks for this suggestion. In the paragraph 2 and the paragraph 3 of the discussion section on page 17, we added the comparison between previous studies and our findings.

  1. Discussion - Future Directions: The discussion lacks suggestions for future research. The authors should discuss the implications of their findings and propose potential directions for future studies.

Reply: Thanks for this suggestion. The suggestions for future research were added in the paragraph 6 of the discussion section on page 18.

  1. Conclusions - Lack of Context: The conclusions do not provide a broader context for the findings. Discussing how the findings could influence the development of therapeutic strategies and future studies would strengthen the conclusion.

Reply: Thanks for this suggestion. We added the context for our findings and discussed that NAD boosters such as NR could be a preventive strategy of insulin resistance in the conclusions section in the page 22.

  1. Overall Structure and Language: The manuscript should be thoroughly proofread to ensure that it is free of typographical errors and that it maintains a coherent and logical flow. Addressing these points will greatly improve the manuscript's clarity, robustness, and impact.

Reply: Thanks for this suggestion. The manuscript was thoroughly and carefully proofread and the errors were corrected to improve the quality of this manuscript.

Round 2

Reviewer 1 Report

None.

Reviewer 2 Report

The manuscript was adequately improved, and the responses are satisfactory. I conclude for accept in present form.